

# Volatile profiling in *Rhus coriaria* fruit (sumac) from three different geographical origins and upon roasting as analyzed via solid-phase microextraction

Mohamed A. Farag[1,2], Nesrin M. Fayek[1] and Ibrahim Abou Reidah[3]

[1] Pharmacognosy Department, College of Pharmacy, Cairo University, Cairo, Egypt
[2] Department of Chemistry, American University in Cairo, New Cairo, Egypt
[3] Department of Chemistry, An-Najah National University, Nablus, Palestine

Corresponding author
Mohamed A. Farag,
mohamed.alifarag@aucegypt.edu

## ABSTRACT

*Rhus coriaria* (sumac) is a fruit grown worldwide for its culinary use as a flavoring agent and for its health benefits. Despite several studies on *R. coriaria* non-volatile metabolites, much less is recognized concerning volatile composition within that genus. In an effort to expand on flavor profile sumac and its food products, we report on volatile profiling from three accessions of different origins including Palestine, Jordan and Egypt in addition to its cold tea and post roasting via headspace solid-phase microextraction (SPME). Under optimized conditions, 74 volatile components were identified belonging to alcohols, aromatics, esters, ethers, furan/aldehyde, hydrocarbons, ketones, monoterpenes, oxides and sesquiterpene hydrocarbons. Major identified components included α-pinene, naphthalene and o-cymene in Palestinian, Jordanian and Egyptian sumac, respectively. Whereas sesquiterpenes amounted for the major volatile class in fresh *R. coriaria* at ca. 40–58%, furan/aldehydes were the predominant classes in roasted fruits (58%). Volatile abundance data was further subjected to multivariate data analyses revealing furfural and nonanal enrichment in roasted compared to fresh fruits and their cold tea preparation. Seeds exhibited no aroma components which justified their removal in *R. coriaria* prior to its use as a food flavor. Such knowledge is expected to be the key for understanding the olfactory and taste properties of *R. coriaria* and its several food products.

## INTRODUCTION

*Rhus coriaria* L., (Family Anacardiaceae) is commonly known as sumac (*Peter, 2012*). Sumac's name is derived from 'sumaga', which simply means red in the Syrian language (*Shabbir, 2012*). Sumac has been traditionally used in many Middle Eastern and Mediterranean countries as a spice, dying agent, and medicinal herb (*Reidel et al., 2017*). It is widely used as a condiment in Turkey and Iran to enhance the taste of poultry and vegetable dishes (*Ravindran, Pillai & Divakaran, 2012*). In Arab countries, sumac is mixed with sesame seeds, salt and thyme in the popular spice mixture called *za'atar* (or *dukkah*).

With regards to its cosmetic use, oils, phytopigments, and proteins derived from the sumac fruit were used in hair care products as anti-dandruff agents, hair colors, and hair cleaning agents, respectively (*Gupta et al., 2010*). Additionally, antioxidants from the sumac fruit were applied to stabilize sunflower oil (*Rayne & Mazza, 2007*). In terms of its folk medicinal use, sumac is reported for treating diarrhea and dermatological problems in addition to reducing blood glucose, uric acid and cholesterol levels (*Candan, 2003*; *Mamedov, Gardner & Craker, 2005*; *Mozaffarian, 2013*). With regards to its health benefits, *R. coriaria* also exerts a myriad of biological effects such as antimicrobial, antiviral, antioxidant, anti-inflammatory, anticancer, hepatoprotective, antihypertensive and cardiovascular protection (*Bozan et al., 2003*; *El Hasasna et al., 2015*; *Pourahmad et al., 2010*; *Rayne & Mazza, 2007*). Bioactive agents reported in *R. coriaria* (*Abu-Reidah, Jamous & Ali-Shtayeh, 2014*) include organic acids, fatty acids, essential and non-essential amino acids, vitamins ($B_1$, $B_2$, $B_6$, $B_{12}$, C, PP), carbohydrates (xylose and glucose), minerals (K, Ca, Mg, Na, P, Fe), tannins, phenolic acids, anthocyanins, flavonoids and terpenoids (*Abu-Reidah et al., 2015*; *Demchik et al., 2015*; *Kossah et al., 2010*; *Kossah et al., 2009*). Although *R. coriaria* is not recognized as an aromatic plant, its fruit is enriched in essential oil composed of monoterpenes and/or sesquiterpenes (*Bahar & Altug, 2009*; *Giovanelli et al., 2017*; *Morshedloo et al., 2017*). Main aroma compounds include nonanal, limonene, 2-decenal, p-anisaldehyde (*Giovanelli et al., 2017*; *Kurucu et al., 1993*), (*E*)-caryophyllene (*Bahar & Altug, 2009*; *Brunke et al., 1993*; *Gharaei et al., 2013*) and the diterpene cembrene (*Gharaei et al., 2013*; *Giovanelli et al., 2017*). Volatile composition in plants is known to be affected by various factors such as geographical origin, harvesting time, processing and agricultural practices (*Morshedloo et al., 2015*). Previous studies have revealed differences in sumac volatile composition as affected by its origin i.e., Turkey, Italy and Iran. Nevertheless, no report has been made on assessing to what extent roasting could affect its aroma profile.

The main objectives of this study were: (1) to assess volatile composition of sumac fruit from different Middle Eastern origins viz. Egypt, Jordan and Palestine using headspace SPME, (2) to assess roasting impact on its aroma profile and (3) to determine sumac cold tea true aroma profile. Volatile abundance data were extracted from chromatograms without prior peak identification in an untargeted manner. Considering the complexity of acquired data, unsupervised and supervised multivariate data analyses viz. principal component analysis (PCA) and orthogonal partial least squares (OPLS), respectively, were employed for classification of fruit samples, and to ensure good analytical rigorousness. To the best of our knowledge, this study provides the first volatile characterization in *R. coriaria* from the Middle Eastern region and report of the impact of roasting on the fruit aroma.

## MATERIALS AND METHODS

### Plant material

*R. coriaria* fruits were collected manually in the full ripe stage from wild trees grown in Nablus, Palestine in October 2016, and were authenticated by Prof. Dr. Ibrahim Abou Reidah, Department of Chemistry, An-Najah National University, Nablus, Palestine. Commercial samples of *R. coriaria* fruits in the full ripe stage were purchased from El

Hen Herbal Company (Amman, Jordan) and Haraz Drugstore (Cairo, Egypt). Voucher specimens were kept at the Department of Pharmacognosy, Faculty of Pharmacy, Cairo University, Egypt. Cold tea of the Palestinian sample was prepared by percolating 10 g of fresh cut fruits (without seeds) in 100 mL distilled water for 10 min, kept at 25 °C, then filtered on Whatman filter paper to remove plant debris. Roasting of the Palestinian sample was carried out by heating fresh cut fruits (without seeds) in an oven set at 120 °C for 20 min. Three replicates were analyzed for each sample. The fruits were stored at −20 °C till further analysis.

## Chemicals and materials

SPME fibers of stableflex coated with divinylbenzene/carboxen/polydimethylsiloxane (DVB/CAR/PDMS, 50/30 μm) (57328-U) or PDMS (polydimethylsiloxane) (57302) were purchased by Supelco (Oakville, ON, Canada). All other chemicals and standards were purchased from Sigma Aldrich (St. Louis, MO, USA).

## Volatile analysis of fresh and roasted fruits

The HS-SPME volatile analysis was carried out as stated previously (*Farag et al., 2017*). Fruits (100 mg) were cut into halves, placed in SPME screw cap vials (1.5 ml) and spiked with ($Z$)-3-hexneyl acetate dissolved in water at a final concentration of 2 μg per vial. The SPME fiber was inserted manually into a vial containing seeds placed in an oven kept at 50 °C for 30 min. The fiber was subsequently withdrawn into the needle and then injected into the injection port of the gas chromatography-mass spectrometer (GC-MS). GC-Ms analysis was performed on a Schimadzu GC-17A gas chromatogram (Schimadzu, Tokyo, Japan) equipped with DB-5 column (30 m × 0.25 mm i.d. × 0.25 μm film thickness; Supelco) and coupled to a Schimadzu QP5050A mass spectrometer. The interface and the injector temperatures were both set at 220 °C. The following gradient temperature program was used for volatile analysis. The oven temperature was kept first at 40 °C for 3 min, then increased to 180 °C at a rate of 12 °C min$^{-1}$, kept at 180 °C for 5 min, and finally ramped up at a rate of 40 °C min$^{-1}$ to 240 °C and kept at this temperature for 5 min. The carrier gas helium was used at a total flow rate of 0.9 mL/min. Splitless injection mode was used for analysis considering the lower levels of volatiles in samples. SPME fiber was prepared for the next analysis by placing it in the injection port for 2 min at 220 °C to ensure complete elution of volatiles. Blank runs were made during sample analyses. The HP quadruple mass spectrometer was operated in EI mode at 70 eV. A scan range was set at *m/z* 40–500.

## GC-MS data processing and multivariate analysis

Volatile components were identified by comparing their retention indices (RI) relative to n-alkanes (C6–C20), mass matching to NIST, Wiley Library Database and with standards whenever available. Peaks were first deconvoluted using AMDIS software (http://www.amdis.net/) prior to mass spectral matching. Volatile abundance data were prepared for multivariate data analysis by extraction using MET-IDEA software (*Broeckling et al., 2006*) for data extraction. Data were then subjected to principal component analysis (PCA), partial least squares-discriminant analysis (OPLS-DA) using SIMCA-P version

13.0 software package (Umetrics, Umea, Sweden). Markers were subsequently identified by analyzing the S-plot, which was declared with covariance (p) and correlation (pcor). All variables were mean centered and scaled to Pareto variance.

## RESULTS

### Volatile analysis of fresh *R. coriaria* fruit (sumac) from three different geographical origins

GC-MS analysis (Tables 1 & 2, Fig. 1) of sumac fruits led to the identification of 74 volatile constituents, categorized in 10 different classes viz. alcohols, aromatics, esters, ethers, furan/aldehydes, hydrocarbons, ketones, monoterpene hydrocarbons, oxide and sesquiterpene hydrocarbons. A typical chromatogram of fresh and roasted sumac fruit aroma profiles is represented in Fig. 1. Initial detection of volatiles started from 0 minutes during chromatographic run but considering that no volatile peaks were detected until 5 min and with only one major peak for acetic acid (Fig. S1), MS detection started from 5 min for all specimens. Considering our interest in volatile terpenoids and hydrocarbons, 5 minutes delay is appropriate for this study. Acetic acid in sumac is likely to derive the tart taste for its fruit. Sesquiterpene hydrocarbons amounted for the most dominant class accounting for ca. 40–58% of the fresh sumac aroma with a total of 26 identified volatile constituents (Table 2). Next to sesquiterpenes, monoterpene hydrocarbons represented the most abundant class (ca. 17–34%) among specimens (Table 2). Other eight volatile classes detected amounted for less than 17% of sumac fruit total volatile blend (Table 2). Naphthalene and α-pinene were the major volatile forms in Jordanian and Palestinian specimens at ca. 15.8 and 16.7%, respectively. Whereas, monoterpene hydrocarbons viz., o-cymene 7.7%, β-ocimene 7.5% and limonene 7.3% were the chief components in fresh sumac fruit aroma derived from Egypt. Volatiles found at comparable levels in all three examined specimens included (*E*)-β-famesene and (*Z*, *Z*)-α-farnesene (sesquiterpene hydrocarbons) detected at ca. 6–8%. With regard to oxides, cineole amounted for 7.3% of Egyptian sumac aroma blend, at two fold the levels that were present in specimens from Jordan (3.2%) and Palestine (2.2%) (Table 1).

In contrast, roasted fruit aroma was predominated by furan/aldehydes at ca. 58% followed by sesquiterpene hydrocarbons at 27%. A dramatic change in fruits' aroma profile was observed upon roasting, exemplified in high furan/aldehyde (58.1%) and ketone (6.1%) levels in roasted fruit concurrent with a marked decrease in the other eight volatile classes (Table 2). Roasted specimens were particularly enriched in furfural (34.3%) and (*E*)-nonanal (12.2%) followed by 3-thujanone (5.1%) (Table 1, Fig. 2). Sumac fruits are also used worldwide to prepare cold tea by simply soaking the fruit in cold water. Consequently, it was of interest to characterize the sumac cold tea aroma profile; a weak aroma profile was detected compared to fruit exemplified by much lower number of volatile components totaling 14 peaks (Fig. 2).

It should be noted that a relatively high standard deviation was observed for some minor constituents (Table 1) viz., *β*-linalool, styrene, octanal, nonanal and β-ocimene especially from the Jordanian specimen. Whether such large variance is associated with the

**Table 1 Relative percentage of volatile compounds detected in fresh *R. coriaria* fruit (Sumac) from three Middle East sites and in response to roasting using SPME-GC–MS measurements (*n* = 3).** The % identified for each class is bolded, while the main component from each site is underlined and bolded.

| Volatile constituents | RT | KI | Fresh sumac | | | | | | Roasted sumac | |
|---|---|---|---|---|---|---|---|---|---|---|
| | | | Egypt | | Jordan | | Palestine | | Palestine | |
| | | | Average (S.D.) | | | | | | | |
| | | | *Alcohols* | | | | | | | |
| **1** | β-Linalool | 9.93 | 1,077 | 3.38 | (1.17) | 1.84 | (1.60) | 0.90 | (0.43) | 0.28 | (0.24) |
| **2** | Endo-Borneol | 11.116 | 1,158 | 0.00 | (0.00) | 0.06 | (0.05) | 0.01 | (0.02) | 0.20 | (0.08) |
| **3** | 4-Terpineol | 11.183 | 1,163 | 0.23 | (0.31) | 0.52 | (0.49) | 0.00 | (0.00) | 0.03 | (0.05) |
| **4** | α-Terpineol | 11.408 | 1,179 | 0.54 | (0.48) | 0.03 | (0.06) | 0.02 | (0.03) | 0.00 | (0.00) |
| **Total alcohols** | | | **4.15** | | **2.46** | | **0.93** | | **0.51** | |
| | | | *Aromatics* | | | | | | | |
| **5** | Styrene | 6.433 | 874 | 0.20 | (0.30) | 0.69 | (0.64) | 1.08 | (0.69) | 0.03 | (0.05) |
| **6** | Naphthalene | 11.328 | 1,176 | 0.00 | (0.00) | **15.88** | (4.18) | 0.00 | (0.00) | 0.00 | (0.00) |
| **Total aromatics** | | | **0.20** | | **16.56** | | **1.08** | | **0.03** | |
| | | | *Esters* | | | | | | | |
| **7** | Methyl nonanoate | 11.608 | 1,193 | 1.30 | (2.08) | 0.26 | (0.25) | 0.00 | (0.00) | 0.10 | (0.18) |
| **8** | Bornyl formate | 11.833 | 1,209 | 0.00 | (0.00) | 0.00 | (0.00) | 0.01 | (0.03) | 0.07 | (0.09) |
| **9** | Linalyl acetate | 11.958 | 1,219 | 0.24 | (0.41) | 0.03 | (0.06) | 0.02 | (0.03) | 0.09 | (0.13) |
| **10** | Bornyl acetate | 12.523 | 1,261 | 0.00 | (0.00) | 0.00 | (0.00) | 0.01 | (0.03) | 0.07 | (0.09) |
| **11** | Isobornyl formate | 12.534 | 1,262 | 0.00 | (0.00) | 0.00 | (0.00) | 0.01 | (0.03) | 0.07 | (0.09) |
| **12** | Nerol acetate | 13.36 | 1,327 | 0.00 | (0.00) | 0.10 | (0.09) | 0.03 | (0.05) | 0.29 | (0.21) |
| **Total esters** | | | **1.54** | | **0.39** | | **0.09** | | **0.68** | |
| | | | *Ethers* | | | | | | | |
| **13** | Estragole | 11.393 | 1,177 | 0.00 | (0.00) | 0.00 | (0.00) | 0.00 | (0.00) | 0.00 | (0.00) |
| **14** | Allyl p-methylbenzyl ether | 13.598 | 1,346 | 4.46 | (1.37) | 0.89 | (0.29) | 2.27 | (2.77) | 0.38 | (0.24) |
| **15** | Precocene I | 14.725 | 1,439 | 1.70 | (2.45) | 0.29 | (0.25) | 0.35 | (0.27) | 0.34 | (0.56) |
| **Total ethers** | | | **6.17** | | **1.17** | | **2.62** | | **0.72** | |
| | | | *Furan/aldehydes* | | | | | | | |
| **16** | Furfural | 5.35 | 822 | 0.32 | (0.34) | 3.73 | (3.26) | 0.00 | (0.00) | **34.37** | (10.55) |
| **17** | Maleic anhydride | 6.092 | 857 | 0.00 | (0.00) | 0.00 | (0.00) | 0.00 | (0.00) | 2.57 | (0.63) |
| **18** | Itaconic anhydride | 7.599 | 935 | 0.90 | (0.53) | 0.29 | (0.38) | 1.80 | (0.57) | 4.29 | (0.48) |
| **19** | Furfural, 5-methyl- | 7.893 | 952 | 0.00 | (0.00) | 0.19 | (0.33) | 0.00 | (0.00) | 3.91 | (1.13) |
| **20** | Octanal | 8.408 | 982 | 1.22 | (0.85) | 0.65 | (0.57) | 0.36 | (0.23) | 0.47 | (0.36) |
| **21** | Nonanal | 10.008 | 1,082 | 1.37 | (1.93) | 0.76 | (0.84) | 0.93 | (0.86) | 12.20 | (5.08) |
| **22** | Decanal | 11.418 | 1,181 | 0.00 | (0.00) | 0.00 | (0.00) | 0.00 | (0.00) | 0.00 | (0.00) |
| **23** | (*Z*)-2-Decenal | 12.204 | 1,237 | 0.18 | (0.26) | 0.55 | (0.47) | 0.40 | (0.19) | 0.38 | (0.46) |
| **Total furan/aldehydes** | | | **3.99** | | **6.17** | | **3.49** | | **58.19** | |
| | | | *Hydrocarbons* | | | | | | | |
| **24** | Dodecane | 11.242 | 1,171 | 0.00 | (0.00) | 0.03 | (0.05) | 0.00 | (0.00) | 0.00 | (0.00) |
| **Total hydrocarbons** | | | **0.00** | | **0.03** | | **0.00** | | **0.00** | |

**Table 1** (*continued*)

| | Volatile constituents | RT | KI | Fresh sumac | | | | | | Roasted sumac | |
|---|---|---|---|---|---|---|---|---|---|---|---|
| | | | | Egypt | | Jordan | | Palestine | | Palestine | |
| | | | | Average (S.D.) | | | | | | | |
| | | | | **Ketones** | | | | | | | |
| 25 | Camphenone, 6- | 9.34 | 1,039 | 1.20 | (1.16) | 0.29 | (0.26) | 0.42 | (0.20) | 0.56 | (0.22) |
| 26 | Acetophenone | 9.549 | 1,052 | 0.18 | (0.15) | 0.00 | (0.00) | 0.01 | (0.02) | 0.00 | (0.00) |
| 27 | 3-Thujanone | 10.105 | 1,088 | 0.93 | (0.59) | 0.89 | (0.77) | 0.84 | (0.07) | 5.16 | (3.74) |
| 28 | Camphor | 10.758 | 1,133 | 0.00 | (0.00) | 0.00 | (0.00) | 0.00 | (0.00) | 0.00 | (0.00) |
| 29 | *p*-Menthone | 10.822 | 1,137 | 0.00 | (0.00) | 0.00 | (0.00) | 0.00 | (0.00) | 0.00 | (0.00) |
| 30 | Pyranone | 10.917 | 1,144 | 0.00 | (0.00) | 0.00 | (0.00) | 0.00 | (0.00) | 0.40 | (0.68) |
| 31 | Carvone | 12.075 | 1,228 | 0.00 | (0.00) | 0.03 | (0.06) | 0.00 | (0.00) | 0.02 | (0.03) |
| **Total ketones** | | | | **2.30** | | **1.21** | | **1.26** | | **6.14** | |
| | | | | **Monoterpene hydrocarbon** | | | | | | | |
| 32 | α-Pinene | 7.143 | 909 | 4.05 | (1.55) | 1.98 | (0.15) | 16.70 | (5.56) | 2.01 | (1.81) |
| 33 | α-Fenchene | 7.45 | 927 | 0.18 | (0.16) | 0.00 | (0.00) | 0.80 | (0.52) | 0.04 | (0.04) |
| 34 | β-Pinene | 7.948 | 957 | 1.79 | (1.46) | 0.50 | (0.44) | 0.84 | (0.52) | 0.11 | (0.06) |
| 35 | β-Myrcene | 7.95 | 956 | 0.88 | (0.54) | 0.18 | (0.16) | 0.17 | (0.29) | 0.17 | (0.12) |
| 36 | β-Thujene | 8.483 | 986 | 1.06 | (0.52) | 0.82 | (0.31) | 1.39 | (1.84) | 0.59 | (0.83) |
| 37 | 4-Carene | 8.602 | 993 | 0.42 | (0.29) | 0.31 | (0.27) | 0.08 | (0.14) | 0.09 | (0.02) |
| 38 | 1,3,8-p-Menthatriene | 8.745 | 1,001 | 1.42 | (0.19) | 0.81 | (0.71) | 0.28 | (0.33) | 0.02 | (0.03) |
| 39 | O-Cymene | 8.768 | 1,003 | 7.73 | (1.27) | 4.25 | (1.87) | 2.40 | (0.82) | 0.23 | (0.09) |
| 40 | Limonene | 8.827 | 1,007 | 7.36 | (3.63) | 3.36 | (0.73) | 3.90 | (1.03) | 0.73 | (0.12) |
| 41 | β-Phellandrene | 8.849 | 1,009 | 0.47 | (0.15) | 0.55 | (0.49) | 0.22 | (0.38) | 0.08 | (0.02) |
| 42 | β-Ocimene | 9.042 | 1,020 | 7.50 | (0.80) | 3.83 | (2.93) | 3.21 | (1.63) | 0.85 | (0.41) |
| 43 | γ-Terpinene | 9.283 | 1,036 | 0.66 | (0.43) | 0.29 | (0.25) | 0.08 | (0.14) | 0.13 | (0.09) |
| 44 | β-Terpinene | 9.289 | 1,035 | 0.33 | (0.15) | 0.17 | (0.16) | 0.06 | (0.11) | 0.19 | (0.04) |
| 45 | Unknown monterpene | 9.7171 | 1,063 | 0.38 | (0.17) | 0.17 | (0.23) | 0.03 | (0.05) | 0.06 | (0.00) |
| 46 | *p*-Cymenene | 9.81 | 1,069 | 0.10 | (0.17) | 0.11 | (0.10) | 0.00 | (0.00) | 0.02 | (0.03) |
| 47 | Unknown monoterpene | 12.533 | 1,262 | 0.00 | (0.00) | 0.00 | (0.00) | 0.15 | (0.15) | 0.18 | (0.20) |
| **Total monoterpene hydrocarbon** | | | | **34.33** | | **17.33** | | **30.32** | | **5.51** | |
| | | | | **Oxides** | | | | | | | |
| 48 | Cineole | 8.908 | 1,012 | 7.27 | (2.98) | 3.25 | (0.81) | 2.23 | (0.15) | 0.53 | (0.31) |
| **Total oxides** | | | | **7.27** | | **3.25** | | **2.23** | | **0.53** | |
| | | | | **Sesquiterpene hydrocarbon** | | | | | | | |
| 49 | Unknown sesquiterpene | 12.631 | 1,270 | 0.00 | (0.00) | 0.13 | (0.11) | 0.01 | (0.02) | 0.06 | (0.08) |
| 50 | α-Longipinene | 13.393 | 1,330 | 0.00 | (0.00) | 0.78 | (0.25) | 0.53 | (0.46) | 0.17 | (0.21) |
| 51 | Copaene | 13.662 | 1,352 | 0.00 | (0.00) | 0.17 | (0.15) | 0.17 | (0.22) | 0.12 | (0.10) |
| 52 | α-Cubebene | 13.675 | 1,353 | 1.29 | (0.40) | 0.60 | (0.07) | 0.93 | (0.67) | 0.27 | (0.22) |
| 53 | Isocaryophyllene | 14.042 | 1,383 | 4.42 | (0.52) | 5.26 | (0.81) | 5.76 | (0.81) | 0.50 | (0.74) |
| 54 | (*E*)-β-Famesene | 14.05 | 1,383 | 6.05 | (1.69) | 6.58 | (1.94) | 8.26 | (1.49) | 0.07 | (0.03) |

| | Volatile constituents | RT | KI | Fresh sumac | | | | | | Roasted sumac | |
|---|---|---|---|---|---|---|---|---|---|---|---|
| | | | | Egypt | | Jordan | | Palestine | | Palestine | |
| | | | | Average (S.D.) | | | | | | | |
| 55 | (Z, Z)-α-Farnesene | 14.119 | 1,389 | 6.02 | (1.98) | 7.62 | (1.97) | 8.60 | (1.41) | 6.84 | (6.42) |
| 56 | Longifolene | 14.2 | 1,396 | 4.42 | (0.52) | 5.23 | (0.81) | 5.73 | (0.84) | 4.81 | (3.96) |
| 57 | Caryophyllene | 14.252 | 1,400 | 4.46 | (1.37) | 5.29 | (1.33) | 6.21 | (1.13) | 4.21 | (3.11) |
| 58 | (E)-β-Famesene isomer | 14.433 | 1,415 | 3.72 | (2.69) | 6.51 | (2.00) | 5.73 | (5.12) | 0.64 | (0.67) |
| 59 | Aromadendrene | 14.558 | 1,425 | 2.15 | (3.04) | 3.70 | (5.72) | 2.74 | (4.10) | 0.45 | (0.30) |
| 60 | Farnesene isomer | 14.575 | 1,426 | 1.29 | (0.40) | 1.75 | (0.15) | 2.81 | (0.97) | 0.38 | (0.42) |
| 61 | α-Humulene | 14.662 | 1,433 | 0.93 | (0.50) | 1.48 | (0.40) | 1.49 | (0.22) | 1.29 | (0.90) |
| 62 | (Z)-Muurola-4(14),5-diene | 14.817 | 1,446 | 1.29 | (0.40) | 0.75 | (0.79) | 1.25 | (0.40) | 0.96 | (0.67) |
| 63 | Germacrene D | 14.841 | 1,449 | 2.83 | (2.38) | 2.46 | (3.56) | 2.34 | (2.71) | 0.82 | (0.61) |
| 64 | γ-Muurolene | 15.083 | 1,468 | 0.00 | (0.00) | 0.27 | (0.24) | 0.51 | (0.79) | 0.60 | (0.17) |
| 65 | α-Muurolene | 15.123 | 1,471 | 0.26 | (0.23) | 0.17 | (0.22) | 0.37 | (0.32) | 0.90 | (0.58) |
| 66 | β-Bisabolene | 15.153 | 1,474 | 0.93 | (0.50) | 1.64 | (0.27) | 1.29 | (1.16) | 0.39 | (0.32) |
| 67 | γ-Cadinene | 15.342 | 1,489 | 0.00 | (0.00) | 0.00 | (0.00) | 0.35 | (0.32) | 0.82 | (0.55) |
| 68 | δ-Cadinene, (+)− | 15.359 | 1,491 | 0.00 | (0.00) | 0.28 | (0.27) | 1.04 | (0.91) | 1.79 | (1.18) |
| 69 | Calamenene | 15.442 | 1,498 | 0.00 | (0.00) | 0.17 | (0.15) | 0.17 | (0.20) | 0.50 | (0.29) |
| 70 | Naphthalene, 1,2,3,4,4a,7-hexahydro-1,6-dimethyl-4-(1-methylethyl)- | 15.575 | 1,507 | 0.00 | (0.00) | 0.17 | (0.15) | 0.68 | (0.61) | 0.36 | (0.24) |
| 71 | Unknown sesquiterpene | 15.628 | 1,511 | 0.00 | (0.00) | 0.41 | (0.40) | 0.98 | (0.86) | 0.30 | (0.19) |
| 72 | α-Calacorene | 15.705 | 1,517 | 0.00 | (0.00) | 0.03 | (0.05) | 0.03 | (0.04) | 0.30 | (0.11) |
| 73 | Unknown sesquiterpene | 15.977 | 1,525 | 0.00 | (0.00) | 0.00 | (0.00) | 0.00 | (0.00) | 0.11 | (0.03) |
| 74 | Unknown sesquiterpene | 21.707 | 1,853 | 0.00 | (0.00) | 0.00 | (0.00) | 0.00 | (0.00) | 0.03 | (0.06) |
| **Total sesquiterpene hydrocarbon** | | | | 40.06 | | 51.44 | | 57.98 | | 27.71 | |
| **Total volatiles** | | | | 100.00 | 0.00 | 100.00 | 0.00 | 100.00 | 0.00 | 100.00 | 0.00 |

experimental setup and detection method has yet to be determined by employing other detectors viz. the flame ionization detector. FID detection or automated SPME in volatiles' extraction step can help minimize such variance.

## Multivariate PCA and OPLS-DA analyses of fresh, roasted and cold tea of *R. coriaria* fruit volatile data

Multivariate PCA (Figs. 3 and 4 & Fig. S2) was carried out to explore the relative variability within the different specimens and to identify geographical origin's (viz. Egypt, Jordan and Palestine) impact on fresh sumac fruit aroma in an untargeted manner. Multivariate data analyses additionally help in identifying potential markers for each fruit origin. Palestinian and Jordanian sumac specimens were found more or less clustered together on the right

**Table 2** Relative percentile (%) of the 10 classes of volatile compounds detected in fresh and roasted *R. coriaria* fruit (sumac) from three Middle East sites as analyzed using SPME-GC–MS.

| No. of volatile constituents | Class | Fresh sumac | | | Roasted sumac |
|---|---|---|---|---|---|
| | | Egypt | Jordan | Palestine | Palestine |
| | | Average (%) | | | |
| 4 | Total alcohols | 4.15 | 2.46 | 0.93 | 0.51 |
| 2 | Total aromatics | 0.20 | 16.56 | 1.08 | 0.03 |
| 6 | Total esters | 1.54 | 0.39 | 0.09 | 0.68 |
| 3 | Total ethers | 6.17 | 1.17 | 2.62 | 0.72 |
| 8 | Total furan/aldehyde | 3.99 | 6.17 | 3.49 | **58.2** |
| 1 | Total hydrocarbons | 0 | 0.03 | 0 | 0 |
| 7 | Total ketones | 2.30 | 1.21 | 1.26 | **6.14** |
| 16 | Total monoterpene Hydrocarbons | **34.33** | **17.33** | **30.3** | 5.51 |
| 1 | Total oxide | 7.27 | 3.25 | 2.23 | 0.53 |
| 26 | Total sesquiterpene hydrocarbon | **40.06** | **51.44** | **58** | **27.7** |

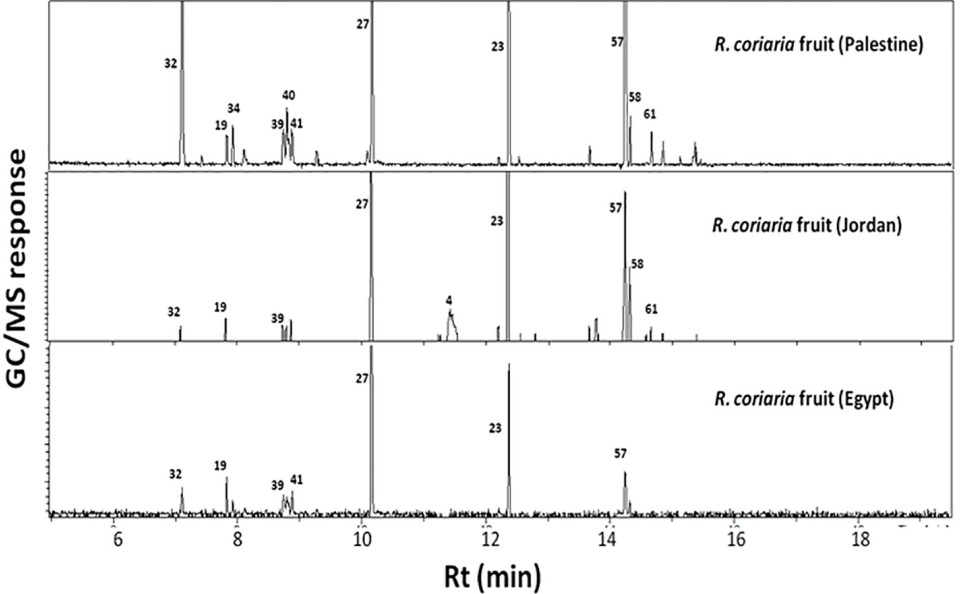

**Figure 1  Representative SPME-GC-MS chromatogram of fresh *R. coriaria* fruit (sumac) collected from Egypt, Jordan and Palestine.** Assigned peak numbers follow those listed in Table 1.

side of PC1 (positive score values). In contrast, Egyptian sumac was positioned on the left side of PC1 (Fig. 3A). A total of 74 volatiles' abundance data were subjected to PCA analysis with two major principle components (PC1/PC2) accounting for 60% of the total variance. A PCA loading plot (Fig. 3B) revealed that α-pinene contributed the most

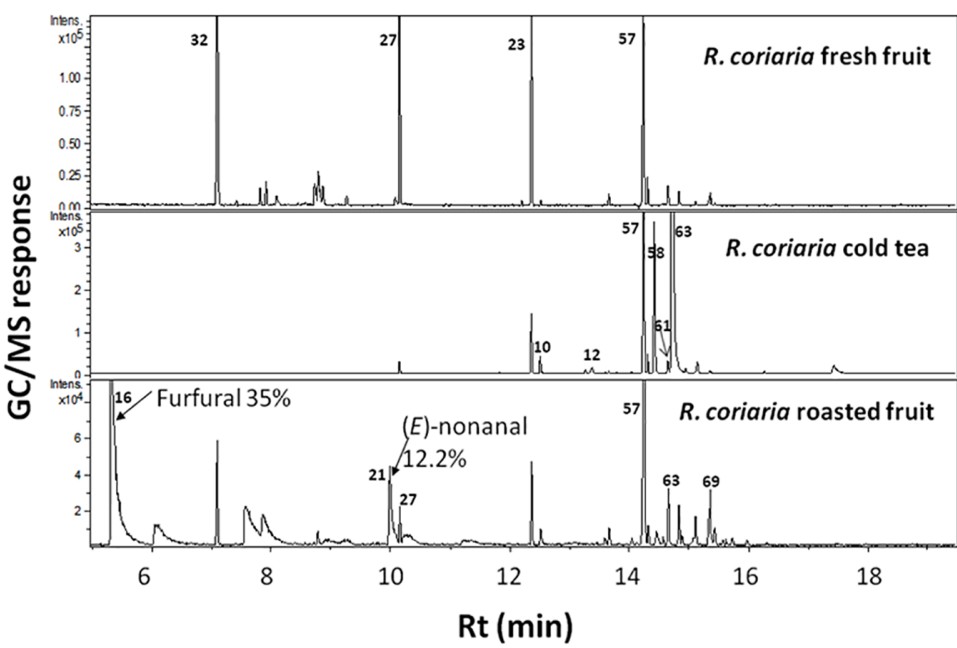

**Figure 2** **Representative SPME-GC-MS chromatogram of fresh, cold tea and roasted *R. coriaria* fruit (sumac) from Palestine.** Assigned peak numbers correspond to volatiles listed in Table 1.

positively along PC1 and PC2, being most abundant in Palestinian sumac in agreement with results presented in Table 1. In contrast, *o*-cymene and limonene located on the far negative side of PC1 were more enriched in Egyptian specimens. Roasting was found to influence sumac aroma profile more than the growth habitat as revealed from PCA analysis (Fig. 4A). Roasted specimens were positioned to the right side of PC1 (positive side) being most distant in composition, whereas fresh fruits and cold tea specimens were all positioned together on the left side of PC1 (negative side). The PCA model (Fig. 4A) was prescribed by PC1 and PC2 accounting for 45% and 24% of the variance, respectively. Unique aroma compounds found in roasted specimens included furfural and nonanal (Fig. 4B). To help identify volatile markers unique for roasted specimens, OPLS-DA (orthogonal projection to latent structures-discriminant analysis) was employed (Fig. S2). The OPLS-DA score plot of roasted versus unroasted fruit showed a clear segregation between roasted and fresh samples explaining 97% of the total variance (R2) and with a prediction goodness parameter Q2 = 95%. The respective S-plot (a remarkable parameter that compares the variable magnitude versus its reliability in OPLS) identified furfural and nonanal as markers of the roasting process (Fig. S2-B) and was in agreement with PCA analysis. The OPLS-DA model of cold sumac tea modelled against fresh fruit failed to provide a fit model as in the roasted specimen case, with lower R2 and Q2 values of 85% and 89% respectively (Fig. S2-C). In general, the aroma of sumac cold tea appeared to be more dominated by esters viz. bornyl acetate, nerol acetate and s esquiterpenes viz. ($\beta$)-caryophyllene and $\alpha$-humulene, as revealed from derived the S-plot (Fig. S2-D).

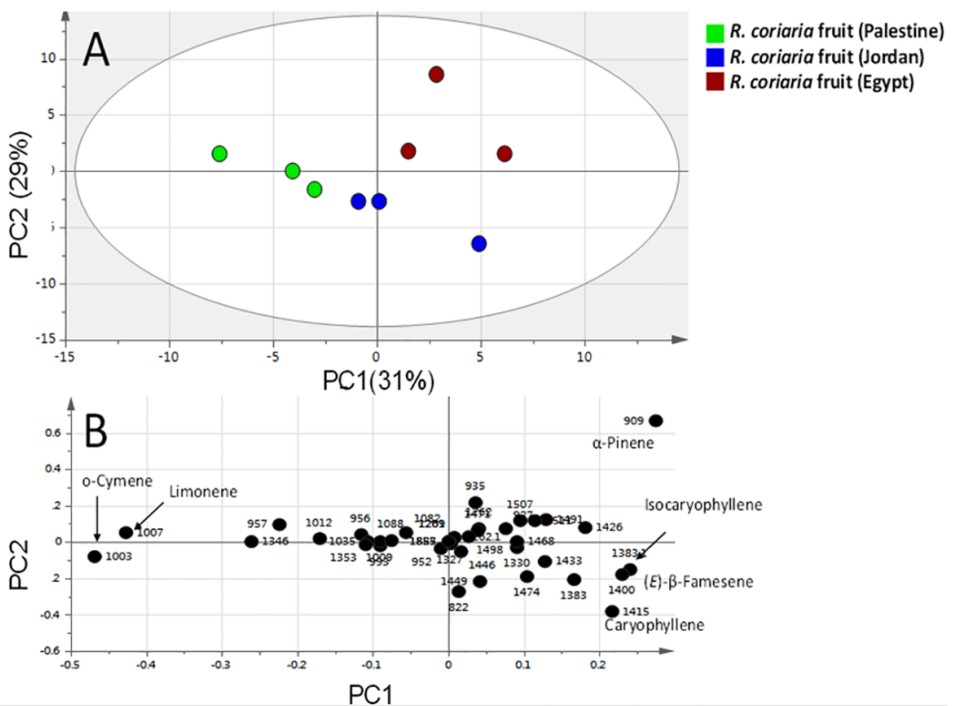

**Figure 3 Principal component analyses of fresh *R. coriaria* fruit (sumac) from three Middle East sites analyzed by SPME-GC–MS ($n = 3$).** Clusters are located at the distinct positions in two-dimensional space described by two vectors of principal component 1 (PC1) = 31% and PC2 = 29%. (A) Score plot of PC1 vs. PC2 scores. (B) Loading plot for PC1 & PC2 contributing volatile peaks and their assignments, with each volatile denoted by its KI value.

## DISCUSSION

Sumac is commonly used as a spice or appetizer, simply by blending its dried ground fruits with freshly cut onion, or mixing it with plant oil or adding it to poultry dishes (*Kossah et al., 2009*; *Shabbir, 2012*) in addition to incorporating it in several nutraceutical products (*Wang & Zhu, 2017*). Also, sumac fruit oil and protein were used in hair care products (*Gupta et al., 2010*).

The main objective of this study was to explore the variation in volatile compositions among *R. coriaria* from three different sites including Palestine, Jordan and Egypt and to assess the impact of roasting on fruits' volatile constituents. With regards to the impact of growth habitat on sumac volatiles' profile, fruit specimens derived from Jordan and Palestine appeared to be similar in volatile composition and in being distant from those of Egypt. Such results are expected considering the close geographical location of Jordan and Palestine and their similar climatic conditions. Abundance of monoterpenes has been reported in sumac fruit grown in Italy (*Giovanelli et al., 2017*; *Reidel et al., 2017*), with α-pinene, β-ocimene and fenchone as the main components. Whereas, prevalence of sesquiterpenes was reported in sumac fruit originated from Turkey and Iran (*Bahar & Altug, 2009*; *Gharaei et al., 2013*; *Morshedloo et al., 2018*) with β-caryophyllene as the most abundant. It should be noted that nonadecane (*Bahar & Altug, 2009*) and p-anisaldehyde

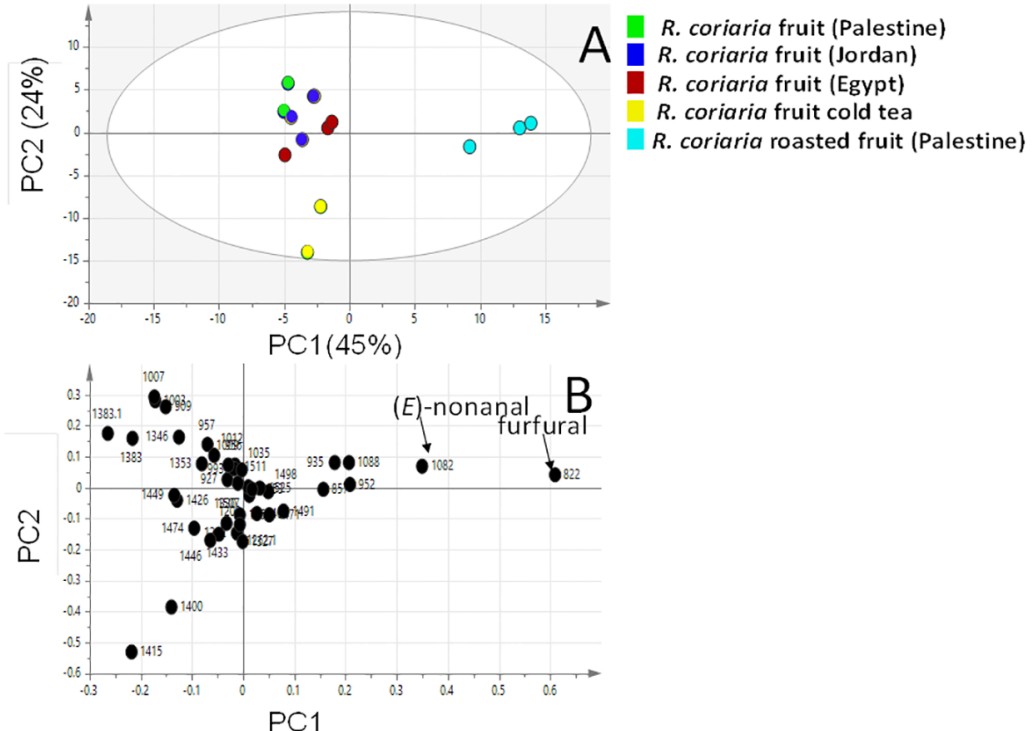

**Figure 4 Principal component analyses of fresh, roasted and cold tea *Rhus coriaria* fruit (sumac) analyzed by SPME-GC–MS ($n = 3$).** Clusters are located at the distinct positions in two-dimensional space described by two vectors of principal component 1 (PC1) = 45% and PC2 = 24%. (A) Score plot of PC1 vs. PC2 scores. (B) Loading plot for PC1 & PC2 contributing mass peaks and their assignments, with each volatile denoted by its KI value.

(*Giovanelli et al., 2017*), previously reported as major components of *R. coriaria* essential oil, were not detected in the current study. Such discrepancy could be attributed to either the different collection methods that are SPME adopted, herein involving no heat treatment in contrary to steam distillation, or to regional differences owing to agricultural or ecological factors (*Bahar & Altug, 2009*; *Giovanelli et al., 2017*).

Upon roasting, a marked variation in volatile profile was detected. This variation is exemplified in higher furfural and nonanal levels in roasted samples. Likewise, previous reports (*Bahar & Altug, 2009*; *Giovanelli et al., 2017*; *Morshedloo et al., 2018*; *Reidel et al., 2017*) on fresh sumac fruit analyzed using SPME from Turkey, Italy and Iran reported furfural and nonanal presence at trace levels, suggesting that these are key markers indicative of the roasting process. Elevated levels of nonanal (23%) were only detected in hydro-distilled sumac fruit in which heating was applied comparable to that of the roasting effect (*Morshedloo et al., 2018*). Our results suggest that both furural and nonanal can be utilized as markers to distinguish between roasted and fresh sumac samples or to predict whether degradation has occurred in sumac fruits upon storage at elevated temperature. Sumac fruits are enriched in both reducing sugar (xylose) and amino acids (*Demchik et al., 2015*), regarded as the precursor compounds for Maillard reaction likely

to occur in sumac fruits upon roasting (*Tamanna & Mahmood, 2015*). Furfurals are major products of Maillard reaction detected in roasted coffee and cocoa beans (*Martins, Jongen & Van Boekel, 2000*), roselle (*Farag, Rasheed & Kamal, 2015*) and during processing of soybeans, pasta and meat (*Tamanna & Mahmood, 2015*). Maillard reaction involves the reaction of amino acids with a reducing sugar in the presence of heat, typical of the roasting process (*Nie et al., 2013*; *Yaylayan, 2006*). Although, furans are of common occurrence in thermal processed foods, increasing awareness of furans' health hazard as a possible carcinogen is recognized (*Nie et al., 2013*; *Reinhard et al., 2004*). According to the FDA guidelines, the average permitted level of furan should not exceed 170 ng/g (*US Food and Drug Administration, 2004*).

In order to identify geographical origin's (viz. Egypt, Jordan and Palestine) impact on fresh sumac fruit aroma profile in an untargeted manner and to help in identifying potential markers for each fruit origin, PCA was attempted to model the volatile abundance data. Palestinian and Jordanian sumac specimens clustered together distant from those of Egyptian sumac. In an attempt to evaluate the effect of heat on sumac, roasting was carried out as previously explained (experimental section). The respective S-plot showed a marked increase in furfural and nonanal upon roasting. Such high furan levels (up to 40%) in roasted specimens warrant more for its quantification in heated sumac food products for safety and health issues.

## CONCLUSION

In the present study, we provide the first comprehensive volatile profile of sumac fruits from three different Middle Eastern countries. A total of 74 different volatile constituents were identified with sesquiterpene hydrocarbons as main class followed by monoterpene hydrocarbons. Egyptian sumac was more enriched in *o*-cymene, β-ocimene and limonene. Whereas, Jordanian and Palestinian specimens exhibited more close volatile profile being enriched in naphthalene and α-pinene. A significant alteration in sumac aroma profile was observed upon roasting, accompanied by a marked increase in furan/aldehydes viz. furfural, 5-methyl furfural, concurrent with a decrease in sesquiterpene and monoterpene hydrocarbons. The prevalence of furans in the roasted sample suggest a distinct change in fruit aroma upon heating and the furan levels should be monitored, considering its health hazardousness. Our volatile profiling provided the true aroma profile in sumac fruit growing in the Middle East, which can be further applied for investigating other factors such as storage, ripening stage, and analyzing its various commercial food products.

### Funding

Mohamed A. Farag was supported by Jesour grant number 30 from the Academy of Scientific Research & Technology (ASRT), Egypt and the American University of Cairo Research Support Grant (RSG1-18). The funders had no role in study design, data collection and analysis, decision to publish, or preparation of the manuscript.

## Grant Disclosures

The following grant information was disclosed by the authors:
Academy of Scientific Research & Technology (ASRT), Egypt: 30.
American University of Cairo Research: RSG1-18.

## Competing Interests

Mohamed A. Farag is an Academic Editor for PeerJ.

## Author Contributions

- Mohamed A. Farag conceived and designed the experiments, performed the experiments, analyzed the data, contributed reagents/materials/analysis tools, prepared figures and/or tables, authored or reviewed drafts of the paper, approved the final draft.
- Nesrin M. Fayek analyzed the data, prepared figures and/or tables, authored or reviewed drafts of the paper, approved the final draft.
- Ibrahim Abou Reidah performed the experiments, contributed reagents/materials/analysis tools.

## Data Availability

The raw data are provided in the Supplemental Files.

## Supplemental Information

Supplemental information for this article can be found online at http://dx.doi.org/10.7717/peerj.5121#supplemental-information.

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
