# Peer review of "Volatile profiling in Rhus coriaria fruit (sumac) from three different geographical origins and upon roasting as analyzed via solid-phase microextraction"

_PeerJ, doi:10.7717/peerj.5121_

## Round 0.1 · original submission · Major Revisions

Thank you for submitting your manuscript to PeerJ. After careful consideration, we feel that the study has merit but does not fully meet PeerJ publication criteria as it currently stands. Therefore, we invite you to submit a revised version of the manuscript that addresses the points raised during the review process.

Reviewer 1 ·

Basic reporting

Dear authors, congratulations for the theme and the work. They have enormous scientific and technological potential. The manuscript is well written, making clear the objectives and results found.

Literature references are pertinent to the content and are well updated, presenting very recent work.

Professional article structure and formatting, apparently conform to the journal's standards. But I suggest that before the definitive publication is reviewed.

Experimental design

In item 2.1 of the Material and Methods, mention that they purchased the fruits in trade and were taken to the department. However, immediately afterwards the following sentence is added: "Cold beverage was prepared by percolating 10 g of powdered fruits in 100 ml distilled water for 10 minutes kept at 25 ° C, then filtered to remove debris".
Thus, it is implied that powdered fruits are purchased fruits that have been made into powder. However, it is not specified in the article how these fruits have reached this stage. I therefore suggest that you add in the Material and Methods how they obtained the powder of the studied fruit.

Validity of the findings

Congratulations on the innovation and originality of the work.

I suggest that the discussion be improved by adding the role of each compound found, where possible, in the quality of the fruit. Not only the nutritional importance, but also the technological importance, especially if this fruit is used as an ingredient in the preparation of some product.

Would this fruit be indicated to be used in some industrialized product to replace synthetic ingredients?

Additional comments

I suggest putting the cat number of the chemicals and reagents used.

I also suggest that in future work the technological potential of this fruit as a natural ingredient in some products is verified.

·

Basic reporting

The title does not reflect the actual work that was performed in this study, the title as is gives the impression that the 3 sumac fruits from different origin was roasted. Suggested modification: “Volatiles profiling in Rhus coriaria fruit (Sumac) from 3 different geographical origins as analyzed via solid-phase microextraction”.

In the abstract section the word flavor was used in the wrong way, once this word is used to describe compounds responsible for taste and for odors, and in the present study it was only performed a volatile profile of the Sumac fruit, not an assessment on its taste.
In the introduction section some comments:
Line 34: how does this fruit is used as flavoring agent in food? As the whole fruit, processed as a powder or essential oil, for example or other form? This was not clear in the text.
Line 35: the reference is very old, Pojero 1891.
Line 49: why the name of the compound cembrene was whiten in quotation marks?
Line 55: the abbreviation for gas chromatography mass spectrometry appears in the text differently, (-) or (/), the authors must check the official/ scientific form of abbreviation.
In general, an improvement/ uptade in the background literature regarding the subject of study is necessary.
In figures 1 and 2 the chromatograms starts with 6 min of analysis, once the analysis was performed by SPME there is no need for solvent delay, so where are the initial compounds. In figure 1, chromatograms of the sumac fruits from Jordan and Egypt show poor resolution; it could be an indication that the SPME conditions could be optimized.
In figure 2, some compounds identified in table 1 as present were not marked in the chromatogram of roasted fruit. For example, alpha-pinene that was identified in the fresh fruit, should have the same retention time in the roasted and was not identified. In addition, as in figure 1 the initial compounds are missing.

Experimental design

The study, especially in the introduction, does not make clear how the knowledge gap is filled. The description of the experimental part is lacking important information, such as:
Line 67: the authors did not explain how the fruits were collected. Once depending on the form species collection, this can influence the sampling and consequently the results. For example, how many units of the species were collected to form the Palestine sample, and how the maturities of the fruits were assessed? Besides the geographical region, degree of maturity is an important factor that could influence the results.
Line 68: the samples from Egypt and Jordan were commercial, this raises the question of how to be sure of the integrity of the collection and degree of maturation of the samples.
Lines 71 and 72: it is not clear how the cold beverage, that latter in the article is called cold tea, was obtained; it was obtained from the fruit powder? How this powder was obtained? What kind of filter was used, with what specifications? And how the filtration was performed, under vacuum?
Line 73: it is not clear how the roasting process was performed? The fruits were whole? And only the Palestine fruits were roasted, this was not clear in the text.
Line 80: in the volatile analysis section, the SPME conditions were not exactly as described in the Farag’s article. In the article cited, the authors used 100 mg of allium sample and a internal standard was used, which did not occur in the present study.
Line 85: the chromatographic conditions were not present.

Validity of the findings

The results as stated need to be revise, and further explanation with literature support is necessary. For example, line 111, alpha-pinene was not the majority compound for Jordan sumac as stated, the majority compound was (Z,Z)-alpha-farnesene according to table 1.
In the results section some errors are present, such as:
Line 103: the word sesquiterpene was misspelled.
Lines 108 and 109: the result on roasted fruit is misplaced between the results for fresh fruit. This sentence should be moved before line 118.

The most serious problem with the results presented has relation to:
- The authors use the GC-MS for assessing relative percentage and this detector has known ionization discrimination, thus the response for different volatile class can vary. The best detector for relative percentage is the flame ionization (FID), which responds to carbon and hydrogen bond or improve analytical care, for example using internal standard to correct response. The authors did not use this FID detector or internal standard.
- Also, the sampling of the fruits from different origin was performed in triplicate (as stated in the experimental section). However, this sampling could be an important point that is making the results not robust enough or statically sound.
- Both of these issues are reflected in the results presented in table 1, several compounds show very high standard deviation, in some cases this deviation is over 50% of the average value, thus making the results not reliable.

In lines 119 and 120, the concentration of some volatile class did not decrease as stated, as shown in table 2, roasting increase the relative percentage of alcohols, although this was not significantly once taking into account the standard deviation. Also, ethers and ester increase, the latter the S.D. is also very high.
In line 133, the PCA analysis is explained wrongly, the sum of both PC, 1 and 2, which is accounted for 60%, not only PC1. Figure 3 has very low significance, once with the sum of PC1 and PC2 only 60% of the data is accounted for.
In the discussion section further comments regarding the reason for the volatile changes in the roasting process is missing, also which are the climate conditions of the fruits from the geographical regions studied, what are their differences, or similarities, also in the other different studies cited.
The authors comment in line 181 that furfural and nonanal could be markers of degradation, what compounds in the fruit are the precursors? This is not discussed. Also, in line 182 the authors state that storage conditions can cause degradation of these fruits, but they do not explain witch conditions, heating, oxygen content, light and etc.
In the conclusion with the errors of the results presented the statement in line 208 is not true.
I wonder the relevance of presenting figure 4, once it brings the results for the fresh fruits from the three geographical origin, together with the roasted from one single origin. The figure 5, presents similar results than figure 4, I suggest exclusion of figure 4.

Additional comments

In general, there are several issues with the data presented by the authors; high standard deviation shows lack of robustness. Experimental conditions are not fully presented. The type of detector chosen should be revised, maybe other experiments or more samples should be analyzed should be conduct to improve data. The background literature of the sumac fruits should be updated and some figures are not needed.

Reviewer 3 ·

Basic reporting

This manuscript presents the volatile profile of Rhus coriaria (Sumac) fruit from three different countries, and compares the effect of roasting on the volatile profile of this fruit. The experiment has a good experimental design and was carried out properly. I recommend that this manuscript should be published after major revisions that were identified among this document, but especially after a language revision from an English native speaker.

The English language should be improved, to increase the quality of the manuscript and ensure a clear comprehension. Some lines present unclear language, including the abstract.
I suggest that you get in touch with some native speaker to correct the English.

Experimental design

No comment

Validity of the findings

No comment

Additional comments

Title: I suggest a few changes in the title of the paper. For example, change the number “3” to the word “three” and also change “upon roasting” that I don’t think is so common to use in the title. One example that I suggest is:

“Volatiles profiling of fresh and roasted Rhus coriaria fruit (Sumac) from three different geographical origins analyzed via solid-phase microextraction”

Keywords: I suggest that you remove the two first ones, since they already appear in the title. Change for the name of the family, or some other popular name of the fruit.

Abstract

Line 23. Standardize the way you express %, first it is close to the number, then the second time it is after a space. Put the first one close to the number as well.

Lines 24-27. Another example of the unclear language, you should try to express the information in a fluid and clear way.

Line 27. “Such knowledge is expected to be the key for understanding the olfactory (…)”. This sentence is vague, which “such knowledge” is that? This is your abstract, you should be very clear and direct in this part, even more when you said that this knowledge is the KEY for understanding something.

Introduction

Lines 33-36. Other example of language that should be improved, the sentences when you introduce the fruit seems truncated.

Lines 36-39. You start both phrases with the same expression “In terms of”. Change one of them.

Line 49. Correct the first “ of the word cembrene.

Line 51. Remove the “or”.

Line 52. Start a new paragraph here, with the objectives of your study.

Line 53. Here and among the manuscript I suggest you to change the word “accessions” to make clearer your meaning.

Lines 53-54. Remove “further” and avoid use “its”, here and among the text because it is not always evident what “its” means. You can change it for “the fruit”or “Sumac” for example.

Lines 54-55. You should write the full names and in the front the acronyms.

Line 56. What do you mean by “untargeted manner”?

Materials and methods

2.1 Plant material – The fruits from Palestine were collected in the full ripe stage, and the fruits from Jordan and Egypt? Describe they stage as well, the ideal is to compare them all in the same stage. Here and among all the manuscript standardize the way you express the temperature, sometimes you write the number together with ºC and sometimes with a space between them.

Line 69. Delete “are”.

Line 71. “in 100 mL OF distilled water”.

Line 72. Add a comma after minutes.

Line 73. “the fruits”

Line 80. Change the / for “and”

Lines 83-84. Remove the ( ).

Results

I suggest you to write the tables and figures already at the beginning of the paragraphs were you discussed them (example: XXX analysis is presented in Table X, Figure X), so the reader can check them and follow the paragraph knowing what they need to see.

Lines 138-141. In this sentence you discussed a very interesting result of your study, but the whole sentence is very confusing. I think that once you get a native English speaker to read and help to improve the language problems you could increase the quality of the description and discussion of the results.

Line 142. Here you could do the suggestion I wrote before, to introduce the figure and what the reader is supposed to find there. It is confusing to read the results in this way.

Discussion

I think that in the first paragraph of the discussion you explained the objectives of your study better than on introduction. I suggest you to insert this part on the end of the introduction and to resume your objective here; since it is the discussion and the reader already saw then on introduction and follow all your manuscript until here.

Line 161. Change “or” for “and”.

Conclusion

Line 209. Delete “or”.

Tables

Standardize the title of the tables. And do not put the description with the title, like you did in Table 1.

Figures

In the figures 3 and 4 you made change that could be very confusing for the reader. In figure 3 you choose the colors from the 3 different regions, the readers saw and possibly spent some time analyzing this figure. Then in figure 4 you change the colors of the already displayed different regions… You should standardize these 3 colors that you already showed to the reader, and change only the additional legends. The fruit from Egypt were green, from Jordan blue and from Palestine red, then on figure 4 Egypt is blue and Jordan yellow, just Palestine that remained red, it could be very confusing for the reader.

Annotated reviews are not available for download in order to protect the identity of reviewers who chose to remain anonymous.

---

## Round 0.2 · Minor Revisions

Please address the remaining concerns of Reviewer 2

Reviewer 1 ·

Basic reporting

The suggested changes were performed and the questions answered, thus the manuscript is ready for publication.

Experimental design

The suggested changes were performed and the questions answered, thus the manuscript is ready for publication.

Validity of the findings

The suggested changes were performed and the questions answered, thus the manuscript is ready for publication.

Additional comments

The suggested changes were performed and the questions answered, thus the manuscript is ready for publication.

·

Basic reporting

No comment

Experimental design

No comment

Validity of the findings

No comment

Additional comments

The authors performed several modifications along the article.
Altought the authors explain the statistical possibility of high values of SD presented in table 01. However, some compounds still have a high standard deviation, to cite a few for example, beta-linalool from Jordan (1.84 ± 1.60); alpha-pinene (2.01 ± 1.81), alpha-fenchene (0.04 ± 0.04), beta-myrcene (0,17 ± 0.12) from roasted fruit. The atuhors stated that they use the present data to compare the samples using multivariate data analyses in nature form and between the roasting process, thus it is important that these SD are as low as possible, even if these SD are consider statistically ok, we as scientists should have a critical assessement of our work and express this concern.
The authors are comparing fruits from different origin based on these values found and in some cases, because of the high SD there are no statistical difference, thus the authors shoul point this as a critic point of their work, and suggest alternatives to improve these results, by using other detector, such as FID, for example, in the end of the discussion. This statement does not invalidate the work.
Also, the authors stated "Considering our interest in volatile terpenoids and hydrocarbons, 5 minutes is appropriate for this study" this is not clear in any part of the article, and it could be important to mention this.

---

## Round 0.3 · accepted · Accept

Your manuscript has been accepted for publication in PeerJ

#